# Time-resolved serial femtosecond crystallography reveals early structural changes in channelrhodopsin

Kazumasa Oda[1†], Takashi Nomura[2†], Takanori Nakane[1†‡], Keitaro Yamashita[1†‡], Keiichi Inoue[3§], Shota Ito[3], Johannes Vierock[4], Kunio Hirata[5,6], Andrés D Maturana[7], Kota Katayama[3], Tatsuya Ikuta[1], Itsuki Ishigami[1], Tamaki Izume[1], Rie Umeda[1], Ryuun Eguma[1], Satomi Oishi[1], Go Kasuya[1#], Takafumi Kato[1], Tsukasa Kusakizako[1], Wataru Shihoya[1], Hiroto Shimada[1], Tomoyuki Takatsuji[1], Mizuki Takemoto[1¶], Reiya Taniguchi[1], Atsuhiro Tomita[1], Ryoki Nakamura[1], Masahiro Fukuda[1], Hirotake Miyauchi[1], Yongchan Lee[1**], Eriko Nango[5,8††], Rie Tanaka[5,8], Tomoyuki Tanaka[5,8], Michihiro Sugahara[5], Tetsunari Kimura[9], Tatsuro Shimamura[8], Takaaki Fujiwara[8], Yasuaki Yamanaka[8], Shigeki Owada[5,10], Yasumasa Joti[5,10], Kensuke Tono[5,10], Ryuichiro Ishitani[1], Shigehiko Hayashi[11], Hideki Kandori[3], Peter Hegemann[4], So Iwata[5,8], Minoru Kubo[2*], Tomohiro Nishizawa[1,6*], Osamu Nureki[1*]

*For correspondence:
minoru@sci.u-hyogo.ac.jp (MK);
t-2438@bs.s.u-tokyo.ac.jp (TN);
nureki@biochem.s.u-tokyo.ac.jp (ON)

[†]These authors contributed equally to this work

Present address: [‡]Structural Studies Division, MRC Laboratory of Molecular Biology, Cambridge Biomedical Campus, Cambridge, United Kingdom; [§]Institute for Solid State Physics, University of Tokyo, Kashiwa, Japan; [#]Division of Integrative Physiology, Department of Physiology, Jichi Medical University, Shimotsuke, Japan; [¶]Preferred Networks, Inc, Tokyo, Japan; [**]Department of Structural Biology, Max Planck Institute of Biophysics, Frankfurt, Germany; [††] Institute of Multidisciplinary Research for Advanced Materials, Tohoku University, Sendai, Japan

[1]Department of Biological Sciences, Graduate School of Science, The University of Tokyo, Tokyo, Japan; [2]Graduate School of Life Science, University of Hyogo, Hyogo, Japan; [3]Graduate School of Engineering, Nagoya Institute of Technology, Nagoya, Japan; [4]Institute of Biology, Experimental Biophysics, Humboldt-Universität zu Berlin, Berlin, Germany; [5]RIKEN SPring-8 Center, Hyogo, Japan; [6]Precursory Research for Embryonic Science and Technology (PRESTO), Japan Science and Technology Agency, Kawaguchi, Japan; [7]Department of Bioengineering Sciences, Graduate School of Bioagricultural Sciences, Nagoya University, Nagoya, Japan; [8]Department of Cell Biology, Graduate School of Medicine, Kyoto University, Kyoto, Japan; [9]Department of Chemistry, Graduate School of Science, Kobe University, Kobe, Japan; [10]Japan Synchrotron Radiation Research Institute, Hyogo, Japan; [11]Department of Chemistry, Graduate School of Science, Kyoto University, Kyoto, Japan

**Abstract** Channelrhodopsins (ChRs) are microbial light-gated ion channels utilized in optogenetics to control neural activity with light . Light absorption causes retinal chromophore isomerization and subsequent protein conformational changes visualized as optically distinguished intermediates, coupled with channel opening and closing. However, the detailed molecular events underlying channel gating remain unknown. We performed time-resolved serial femtosecond crystallographic analyses of ChR by using an X-ray free electron laser, which revealed conformational changes following photoactivation. The isomerized retinal adopts a twisted conformation and shifts toward the putative internal proton donor residues, consequently inducing an outward shift of TM3, as well as a local deformation in TM7. These early conformational changes in the pore-forming helices should be the triggers that lead to opening of the ion conducting pore.

# Introduction

Channelrhodopsins (ChRs) are light-gated cation channels first identified as sensory photoreceptors in green algae (*Nagel et al., 2003*). ChRs consist of seven transmembrane helices and a retinal chromophore covalently attached to a conserved lysine residue via a Schiff base linkage. ChRs allow fast ion flux across cell membranes upon illumination and thus are widely utilized as optogenetic tools to precisely control neural activity (*Schneider et al., 2015*; *Ernst et al., 2014*; *Zhang et al., 2011*). As in the well-studied bacterial proton pump bacteriorhodopsin (BR), light absorption triggers all-*trans* to 13-*cis* isomerization of the retinal, and induces the transitions of optically distinguishable intermediates that lead to channel opening and closing (photocycle process) (*Kuhne et al., 2019*; *Bamann et al., 2010*; *Bamann et al., 2008*; *Figure 1*). Spectroscopic studies have provided detailed characterizations of the respective intermediates in the photocycle of ChRs. In the best studied variant, *Chlamydomonas reinhardtii* ChR2 (*Cr*ChR2) (*Bamann et al., 2008*; *Stehfest and Hegemann, 2010*; *Lórenz-Fonfría et al., 2013*), retinal isomerization leads to an intermediate with a red-shifted absorption peak, $P_1^{520}$ (K or L in BR), and sequential deprotonation and reprotonation of the Schiff base generate $P_2^{390}$ and $P_3^{520}$ (M and N in BR), respectively (*Figure 1*). Photocycle transitions are determined in most ChRs by an aspartate–cysteine pair in the retinal binding pocket (DC-pair) (*Berndt et al., 2009*; *Yizhar et al., 2011*; *Figure 1—figure supplement 1*), but the mechanism by which ion pore formation is associated with these residues is still under debate. Recent studies have suggested that the ion conducting pore is formed during the late $P_2^{390}$ intermediate *Kuhne et al., 2019*, involving the rearrangement of the pore-forming TM helices (*Müller et al., 2011*), and a two-dimensional crystal study indeed revealed low resolution projection maps of the light-induced helix movement in ChR (*Müller et al., 2011*; *Müller et al., 2015*). A detailed mechanism for how retinal isomerization causes conformational changes and how these changes induce ion pore opening has remained elusive.

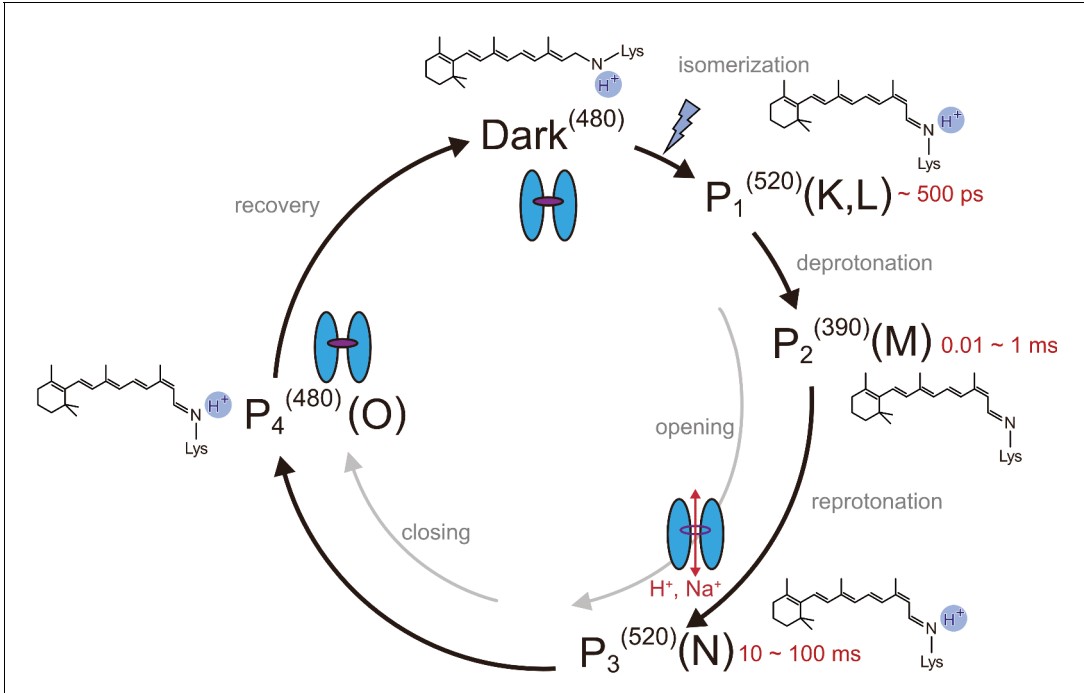

**Figure 1.** Photocycle model of channelrhodopsin. Schematic model of the C1C2 photocycle. Superscript of each reaction intermediate ($P_1^{(520)}$, $P_2^{(390)}$, $P_3^{(520)}$, $P_4^{(480)}$) indicates the wavelength of maximum absorption. Open/close states of the channel pore and the proposed model of the retinal and protonation of Schiff base are indicated. The time range in which the $P_1$–$P_3$ intermediates were observed in the spectroscopic studies in solution is shown.

The online version of this article includes the following figure supplement(s) for figure 1:

**Figure supplement 1.** Sequence alignment of channelrhodopsin (ChR) variants.

The recent development of the X-ray free electron laser (XFEL) technology has enabled high-resolution visualization of the time-resolved molecular conformational changes in crystals, thus profoundly advancing our understanding of biological reactions (*Tosha et al., 2017*; *Nango et al., 2016*; *Suga et al., 2017*; *Shimada et al., 2017*). In this study, we performed the time-resolved serial femtosecond crystallography (TR-SFX) analysis of ChR, which revealed critical conformational changes occurring early in the photocycle that probably induce the cation permeation in the later stages.

## Results

### Time-resolved spectroscopy of C1C2

We used a chimeric construct between *Chlamydomonas reinhardtii* ChR1 (*Cr*ChR1) and ChR2 (*Cr*ChR2) for the TR-SFX experiments (hereafter referred to as C1C2) (*Kato et al., 2012*; *Figure 1—figure supplement 1*), which has normal ChR1 photocycle properties (*Hontani et al., 2017*; *Inaguma et al., 2015*; *Figure 2a,b*). C1C2 yielded high-density microcrystals that diffracted XFEL to about 2.3 Å at SACLA (*Figure 3a,b*). The dark state structure of C1C2 determined at SACLA is almost the same as the previous structure solved by the synchrotron analysis, including the

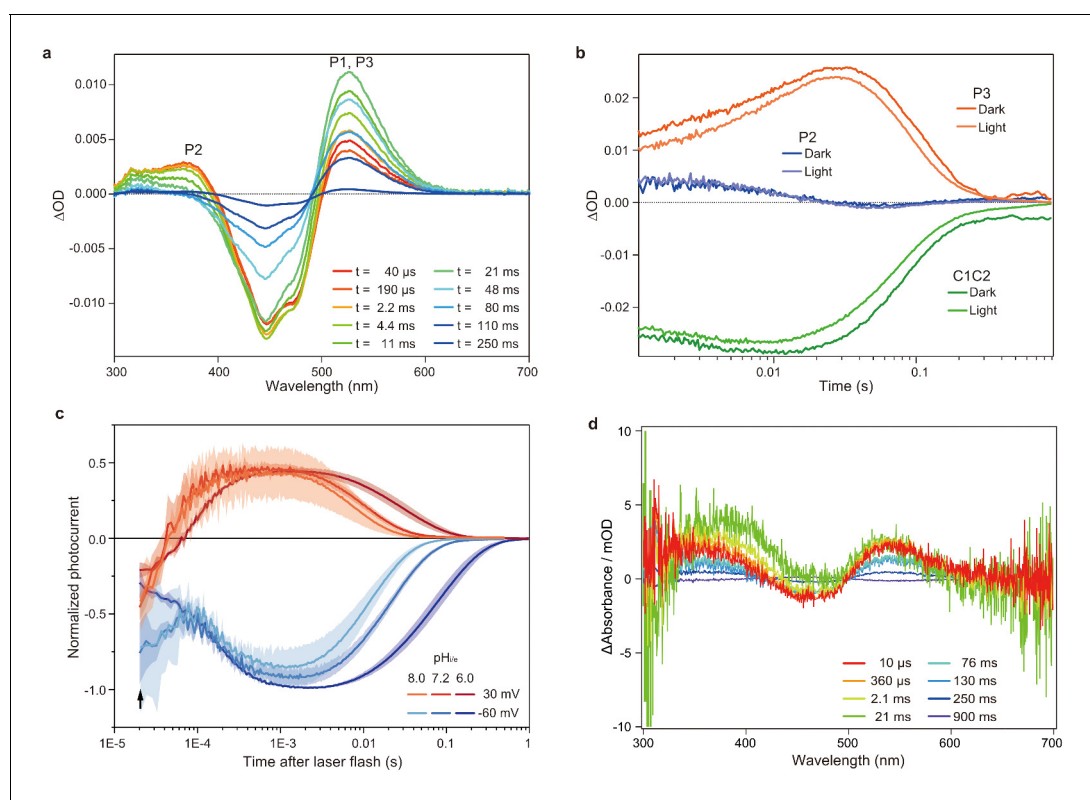

**Figure 2.** Flash photolysis and flash photo activation measurement of C1C2. (**a**) Transient absorption spectra of C1C2 reconstituted in POPE/POPG (protein/lipid molar ratio = 1/50), 150 mM NaCl, 50 mM Tris-HCl (pH 8.0), 5% glycerol, and 0.01% cholesteryl hemisuccinate (CHS). (**b**) Time trace of absorption changes of C1C2 reconstituted in POPE/POPG, at 445 (green), 375 (blue), and 520 (red) nm probe wavelengths, which roughly estimate C1C2 (ground state), P2, and P3 formation, respectively. (**c**) Normalized, log-binned and averaged photocurrents of the C1C2 protein (mean ± SEM, n = 3 - 5). The black arrow indicates the inward directed current caused by retinal Schiff base deprotonation. (**d**) Transient difference absorption spectra recorded from the multiple C1C2 crystals. Each time-resolved spectrum was averaged over 21 crystal data sets. The time evolution is indicated from red to purple.

The online version of this article includes the following figure supplement(s) for figure 2:

**Figure supplement 1.** Flash photolysis measurements of C1C2 solution in n-dodecyl-β-D-maltoside (DDM).

**Figure supplement 2.** Voltage-clamp recordings in HEK293 cells of photocurrents from C1C2.

**Figure supplement 3.** Flash photolysis measurements of C1C2 crystals and C1C2 solution in n-dodecyl-β-D-maltoside (DDM).

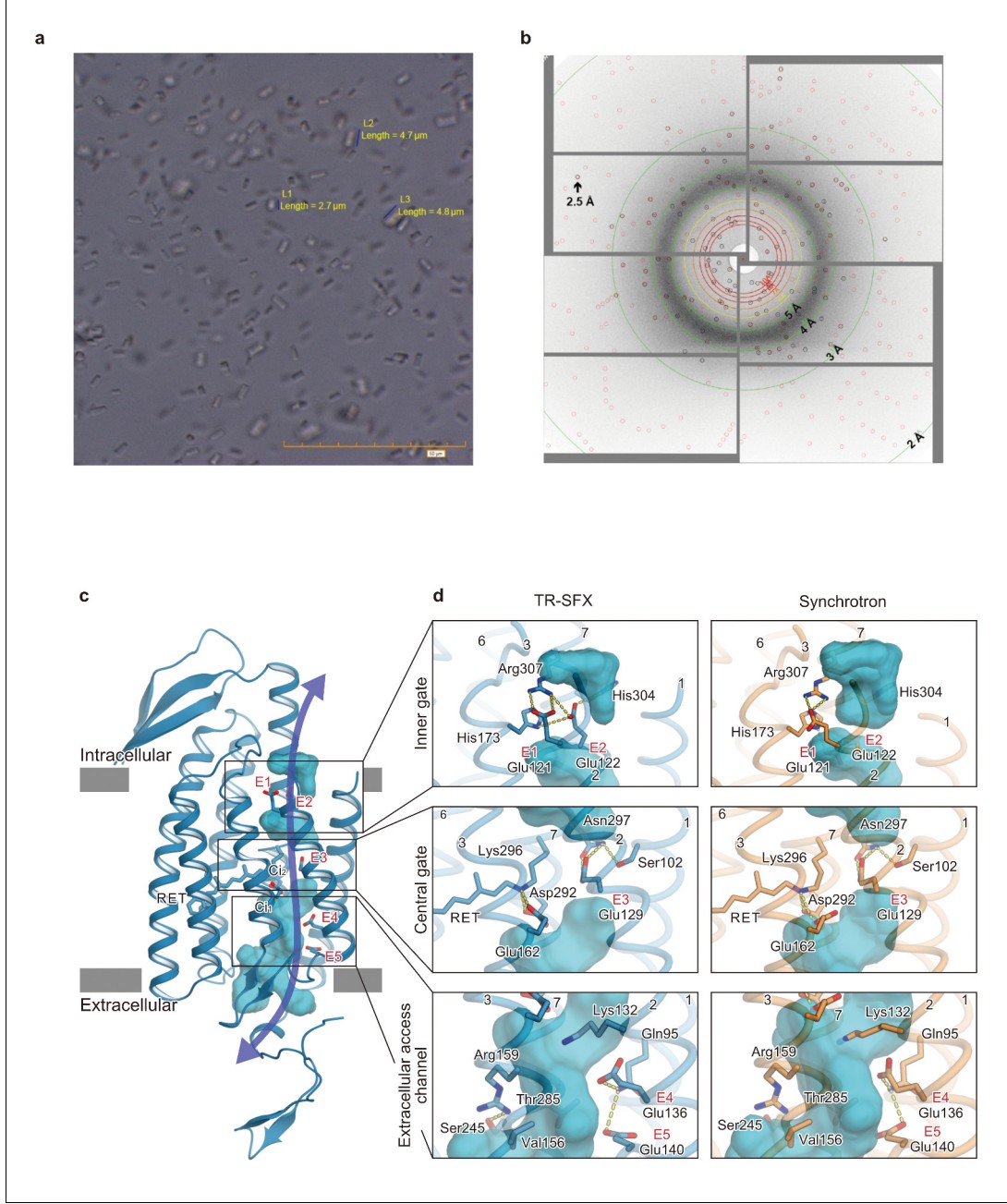

**Figure 3.** Microcrystals for serial femtosecond crystallography (SFX) experiment and C1C2 SFX structure. (**a**) Lipidic cubic phase (LCP) crystals of C1C2 optimized for the time-resolved SFX (TR-SFX) experiments. The orange scale bar on the lower right indicates 50 μm, with 5 μm sub-scaling lines. The size of the crystals ranged from 2 to 5 μm. (**b**) A diffraction image from a C1C2 crystal, obtained with a single SACLA-XFEL pulse. (**c**) The structure of dark state C1C2 determined by serial femtosecond crystallography. A water accessible cavity is illustrated, with the putative ion pathway indicated by an arrow. The five glutamic acid residues lining the ion pore (E1–5) and the two counterion residues ($C_{i1}$ and $C_{i2}$) are indicated by sticks, and the three constriction sites, the inner, central, and outer gates, are enclosed in boxes. (**d**) Comparisons of the constriction sites of the TR-SFX structure (left panels) and the synchrotron structure (right panels; PDB code 3UG9) of C1C2, for the inner (upper panels), central (middle panels), and outer (lower panels) gates. The constituent residues are shown as sticks, and the TM helix number is indicated on each helix.

The online version of this article includes the following figure supplement(s) for figure 3:

**Figure supplement 1.** Electron density of retinal.

**Figure supplement 2.** Comparison of the crystal packing between C1C2 and *Cr*ChR2.

*Figure 3 continued on next page*

*Figure 3 continued*

**Figure supplement 3.** Schematic model of the time-resolved serial femtosecond crystallography setup.

chromophore retinal, the interhelical interactions of the inner and central gates, and the extracellular water access channel in the dark state (*Figure 3c,d*, and *Figure 3—figure supplement 1*; *Hontani et al., 2017*). The C1C2 photocycle was reported in detail recently (*Hontani et al., 2017*), revealing different accumulations of the $P_1$ and $P_2$ photo-intermediates, as compared to *Cr*ChR2. Preceding the TR-SFX experiment, we investigated the photocycle and ion pore formation processes of C1C2. In the flash-photolysis experiments, C1C2 showed similar photo-intermediates as observed in the previous study (*Hontani et al., 2017*), in detergent micelles or reconstituted lipid membranes (*Figure 2a,b* and *Figure 2—figure supplement 1*). The results indicate a rapid rise of 520 nm absorption after photo-excitation, occurring before 10 µs, which corresponds to the formation of the $P_1$ intermediate with an isomerized retinal (*Hontani et al., 2017*; *Nogly et al., 2018*), and the subsequent increases at 390 nm and 520 nm show formation of the $P_2$ and $P_3$ intermediates with a deprotonated and reprotonated retinal Schiff base, respectively (*Figure 2a*). The absorption spectra of $P_1$ and $P_3$ resemble each other and cannot be distinguished in C1C2, but the delayed rise of the 530 nm absorption after 360 µs shows the formation of the $P_3$ intermediate at the later time point. To correlate the different photocycle intermediates to formation of the ion conducting pore, we next measured the C1C2 photocurrents evoked by 7-ns laser flash (*Figure 2c* and *Figure 2—figure supplement 2*). The C1C2 photocurrents are initiated by a fast, inward directed charge transfer that likely coincides with the retinal deprotonation ($P_2$ formation), which has not been fully time-resolved in the whole cell patch clamp recordings. Afterwards, passive photocurrents rise with a time constant of 70–140 µs (rather faster at acidic pH) and change direction depending on the membrane voltage. The photocurrent remains after reprotonation of the retinal Schiff base and declines in a biexponential manner, with two voltage dependent time constants of 8–40 ms and 60–130 ms and a kinetic amplitude ratio that depends on the pH, correlating neither precisley with the $P_2$ nor the $P_3$ intermediate. The absorption spectrum of ChR depends mostly on the microenvironment of the chromophore retinal, especially on the Schiff base environment (*Ritter et al., 2008*), and does not need to reflect changes in the ion pore structure . As recently shown for *Cr*ChR2 (*Kuhne et al., 2019*), also C1C2 channel opening takes place only after deprotonation of the retinal Schiff base, during further confirmational changes of the $P_2$ intermediate.

To determine the time points for the TR-SFX study, we next investigated the photocycle of C1C2 in the lipidic cubic phase (LCP) crystals. The flash illumination induced a rapid elevation at 530 nm, confirming retinal isomerization ($P_1$ formation) (*Figure 2d* and *Figure 2—figure supplement 3a, c*). The elevation of the 390 nm absorption, which is most prominent in 2.1–21 ms, showed the $P_2$ formation, but the subsequent increase at the 530 nm absorption was not observed in the crystal (*Figure 2d* and *Figure 2—figure supplement 1* and *Figure 2—figure supplement 3a,c*), suggesting that the transition from $P_2$ to $P_3$ is hindered in the crystal, although the C1C2 crystal has only minimal interactions at the extracellular membrane region (*Figure 3—figure supplement 2*). Nonetheless, the accumulation of the $P_2$ intermediate is more prominent in the crystals, as the spectral increase at 390 nm is relatively higher, as compared to the similar increase detected in micelles and reconstituted membranes (*Figure 2a,d* and *Figure 2—figure supplement 1*, *Figure 2—figure supplement 3b,d*). In addition, the flash photo-activation experiment using HEK cells showed that the photocurrent rose in between 100–1000 µs, regardless of pH, which corresponds to a second late $P_2$ intermediate (*Figure 2c* and *Figure 2—figure supplement 2*), in a good agreement with the previous studies in ChR2 (*Kuhne et al., 2015*; *Kuhne et al., 2019*). Therefore, we selected the time points corresponding to the $P_1$ to $P_2$ transition in the crystals for the TR-SFX study, to understand the detailed mechanism of the light-dependent channel opening in ChR.

## Structural change in ChR

The LCP matrix containing C1C2 microcrystals was introduced to the XFEL chamber in a continuous flow. Since the crystallized sample was swollen and almost in the liquid sponge phase, we added paraffin and solid LCP matrix to enable stable and constant sample flow to the FEL beam (detailed in Materials and methods). The diffraction images were collected at 30 Hz, with the 470 nm pump-

pulse laser at 15 Hz and the delayed time points at 1 µs, 50 µs, 250 µs, 1 ms, and 4 ms (*Figure 3—figure supplement 3a,c* and *Supplementry file 1*). The absence of light contamination was checked by the second dark data set, obtained with a 10 Hz pump laser pulse (*Figure 3—figure supplement 3c,d*). The difference-Fourier maps were calculated at 2.5 Å for each delayed time point, with the dark state SFX structure as the reference (*Figure 4*), and they clearly revealed the structural changes in the protein core region of C1C2 (*Figure 4—figure supplement 1*; *Figure 4—figure supplement 2* and *Video 1*). Due to the low quantum efficiency of the retinal isomerization, which is estimated as only about 30% in C1C2 (*Hontani et al., 2017*), we calculated the extrapolated structure factor amplitudes and the models were refined against the extrapolated maps (detailed in Materials and methods and *Figure 4—figure supplement 3*). We also calculated the $F_c^{light}$–$F_c^{dark}$ maps using the models, which shows a similar appearance with the difference-Fourier map and thus confirms the conformational change (*Figure 4—figure supplement 4*).

Since the retinal isomerization reaction is generally completed within a few picoseconds (*Hontani et al., 2017*; *Neumann-Verhoefen et al., 2013*; *Bühl et al., 2018*), the observed changes correspond to the events following the retinal isomerization. The isomerization of the retinal induces conformational changes of the retinal-linked residue, Lys296, and TM7 (*Figure 4*, left panels). The positive densities along Lys296 suggest its downward shift (toward the extracellular side), which is detected from 1 µs and becomes most prominent at 4 ms (*Figure 4*, left panels). In accordance with this shift, positive and negative difference densities are observed around the proximal residues in TM7, such as Trp299, indicating an overall downward shift of the middle portion of TM7. The similar downward shift of TM7 (Helix G) was the most distinct feature in the L-like intermediate of BR, observed in both the cold-trapped synchrotron structure (*Neutze et al., 2002*) and the recent TR-SFX study (*Nango et al., 2016*). In the TR-SFX experiment of BR, the primary retinal isomerization reaction induced local distortion around the $C_{13}$–$C_{14}$ double bond of the retinal, which was subsequently propagated as conformational changes to other protein regions, also involving a downward shift of TM7. Therefore, the retinal-attached residue in C1C2 probably undergoes similar conformational changes, although the primary retinal isomerization was not visualized, due to the limited resolution of the current TR-SFX experiment. Notably, at 50 µs we observed positive and negative difference densities at the putative internal proton acceptor residue, Asp292 (*Figure 4*, left panels), indicating a slight downward shift of this residue, which might be associated with the proton accepting reaction.

## Retinal kink-induced outward shift of TM3

The most prominent change occurs in TM3 (*Figure 4*, right panels). A strong negative density appears at Cys167, accompanied by a complementary positive density at the distal site from the retinal, suggesting an outward shift of Cys167. In addition, a strong positive density appears near Pro168 on the outside of the TM3 segment, which indicates an outward shift of the cytoplasmic segment of TM3, originating at the helix kink at the Pro168 residue. Along with these changes, negative and positive densities appear at the $C_{13}$ methyl group and on the lateral side of the polyene chain of the retinal, respectively. This kink and lateral shift of retinal allows lateral movement of TM6 as well as TM3 (*Figure 4*, right panels). The current difference Fourier map is in excellent agreement with the recent MD simulation combined with QM/MM calculations (*Cheng et al., 2018*), in which the retinal adopts a kinked conformation toward TM3 upon the Schiff base deprotonation, probably due to the destabilized resonance structure of the polyene chain (*Figure 5*), although the QM/MM simulation focuses only on the molecular events after the Schiff base deprotonation and thus does not exactly match the experimental time scale. Together, the sequential changes in the difference-Fourier maps suggest that the retinal kink occurs, followed by the consequent outward shift of TM3. Consistently, the positive/negative peaks at the retinal appear slightly earlier, whereas the change in TM3 becomes slower and reaches the maximum at the later time points of 1 ms and 4 ms (*Figure 5*).

The current TR-SFX experiment revealed two critical conformational changes occurring early in the photocycle process: the outward shift of TM3 and the downward shift of TM7, both of which constitute gate constriction sites along the ion permeation pathway in the dark state (*Figure 3*). TM7 contributes essentially to the 'central gate', in which Asn297 forms hydrogen bond interactions with Ser102 (TM1) and Glu129 (TM2). Therefore, the TM7 shift, as well as the outward shift of Asp292 upon proton acceptance, should lead to rearrangements of the central gate residues. The

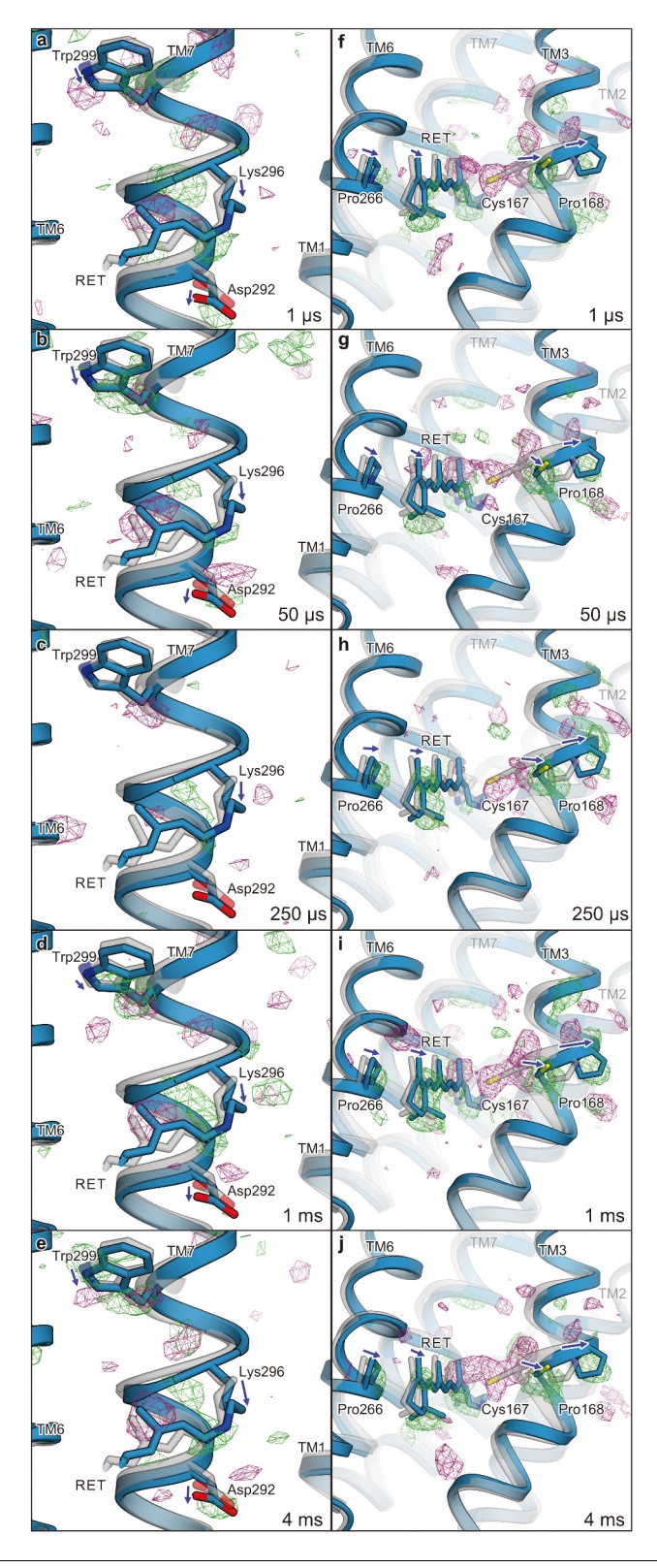

**Figure 4.** Difference Fourier electron density map and structural changes around TM7 and TM3. Views of the |F$_{obs}$|$^{light}$− |F$_{obs}$|$^{dark}$ difference Fourier electron density maps and the structural changes around TM7 (**a–e**) and TM3 (**f–j**) for 1 μs (**a and f**), 50 μs (**b and g**), 250 μs (**c and h**), 1 ms (**d and i**), and 4 ms (**e and j**). Green and purple meshes indicate positive and negative difference electron densities, respectively (contoured at ±3.1 to 3.3σ, see also

*Figure 4 continued on next page*

*Figure 4 continued*

*Figure 4—figure supplement 1*). The difference Fourier maps were calculated by using the phases from the coordinates of the dark-adapted C1C2 structure (gray). Paired negative and positive difference electron densities indicate the downward shift of TM7 and the outward shift of TM3. The structural model (blue) was refined against the extrapolated data of each time point and the model was superimposed upon the initial-state C1C2 model (gray). Movements of the residues and the TM helices are indicated by arrows. Probably due to the variations in the crystal quality (Table 1), the difference density in the 250 µs time-delay seems rather weak.

The online version of this article includes the following figure supplement(s) for figure 4:

**Figure supplement 1.** Overview of the $|F_{obs}|^{light} - |F_{obs}|^{dark}$ difference Fourier electron density maps for five time points.

**Figure supplement 2.** Stereo views of the difference Fourier electron density maps and structural changes.

**Figure supplement 3.** Electron density of extrapolated map.

**Figure supplement 4.** Calculated difference $|F_{calc}|^{light} - |F_{calc}|^{dark}$ map and structural changes around TM7 and TM3.

**Figure supplement 5.** Difference Fourier maps along the putative ion pore.

intracellular end of TM3 shapes the 'inner-gate', in which His173 forms ionic interactions with Arg307 in TM7 and Glu121 and Glu122 in TM2. The recent MD simulation showed a similar retinal kink and subsequent water influx from the intracellular side (*Figure 5*; *Cheng et al., 2018*). Therefore, the conformational change in the middle of TM3 might propagate to the intracellular segment and induce the opening of the inner gate.

Cys167 and Asp195 have been identified as important residues for channel gating and are thus termed DC-gate or DC-pair, which is conserved among most ChRs. FTIR and structural studies have suggested that Asp195 functions as the internal proton donor for the deprotonated Schiff base, via an internal hydrated water molecule and Cys167 (*Nack et al., 2010*; *Bamann et al., 2010*; *Lórenz-Fonfría et al., 2013*). A previous FTIR study showed an increase of the S-H stretching frequency of the Cys167 side chain upon illumination (*Ito et al., 2014*). The DC-pair residues are critical for the fast channel kinetics of ChRs, as mutations of any of them drastically slowed the open/close kinetics of the channel, resulting in step function opsin (SFO) mutants, with long-lasting photocurrents that can be switched on and off by different wavelengths of light (*Berndt et al., 2009*; *Yizhar et al., 2011*). Previous 2D crystallographic studies of the *Cr*ChR2 C128T mutant (corresponding to Cys167 in C1C2 [*Figure 1—figure supplement 1*]) suggested rearrangements of TM2, 6, and 7 in the open channel conformation, that was here induced by the stationary light (*Müller et al., 2011*; *Müller et al., 2015*). These conformational changes are partly consistent with the current TR-SFX analysis. However, the TM3 shift is only observed in the current study, indicating the critical role of the Cys residue in the TM3 shift. The movement of TM7, on the other hand, is similarly observed in both experiments, although to different degrees. We therefore propose that the conformational changes in TM7 and TM3 are the

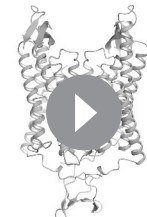

**Video 1.** Overview of the $|F_{obs}|^{light} - |F_{obs}|^{dark}$ difference. Fourier electron density maps for five time points and structural changes. Views of the $|F_{obs}|^{light} - |F_{obs}|^{dark}$ difference Fourier electron density maps and the structural changes. Green and purple meshes indicate positive and negative difference electron densities, respectively (contoured at ±3.1 to 3.3σ, see also *Figure 4—figure supplement 1*). The difference Fourier maps were calculated by using the phases from the coordinates of the dark-adapted C1C2 structure (gray). Paired negative and positive difference electron densities indicate the downward shift of TM7 and the outward shift of TM3. The structural model (blue) was refined against the extrapolated data of each time point and the model was superimposed upon the initial-state C1C2 model (gray). Movements of the residues and the TM helices are indicated by arrows. Probably due to the variations in the crystal quality (*Supplementary file 1*), the difference density in the 250 µs time-delay seems rather weak.

https://elifesciences.org/articles/62389#video1

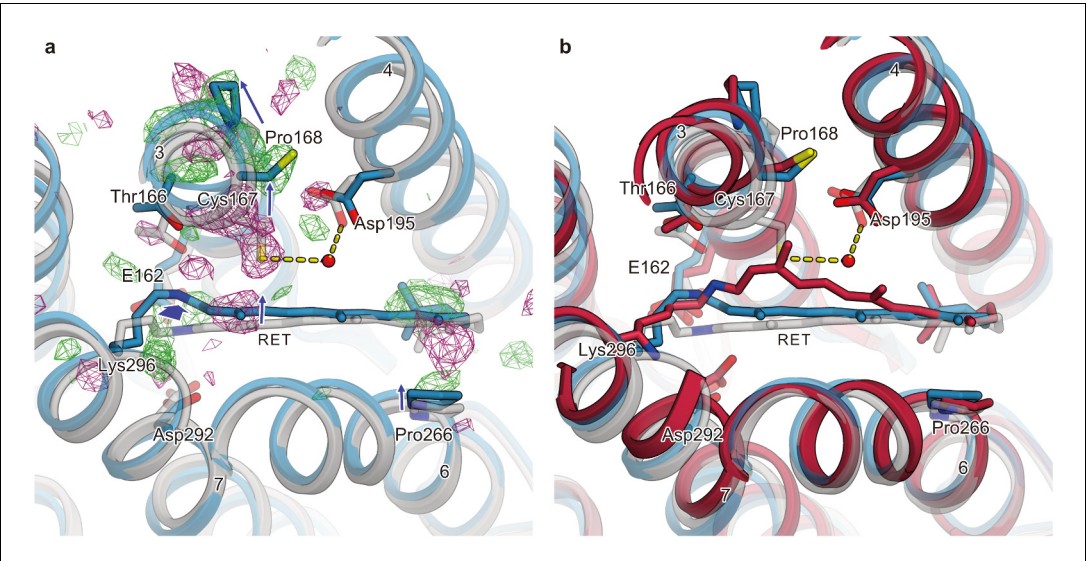

**Figure 5.** Conformational change of retinal and comparison with QM/MM model. (**a**) Intracellular view of the |$F_{obs}|^{light}$ − |$F_{obs}|^{dark}$ difference Fourier electron density map and the structural changes around the retinal. Green and purple meshes indicate positive and negative difference electron densities, respectively. The structural model (blue) was refined against the extrapolated data of 1 ms and the model was superimposed upon the initial-state C1C2 model (gray). Movements of residues or TM helices are indicated by arrows. (**b**) The retinal pocket of dark-adapted C1C2 (gray), the model at the 1 ms delayed-time point (blue), and the model at 400 ns of the QM/MM simulation (*Cheng et al., 2018*) (red) are superimposed together.

two important triggers that cooperatively induce channel opening by eliciting sequential conformational changes in the surrounding helices, such as TM2 and TM6 (*Figure 6*), as observed in the 'open' conformation of the SFO mutant of *Cr*ChRs (*Müller et al., 2015*). This model is also in good agreement with the results of the vibrational spectroscopic study showing a two-phasic helix hydration occurring during the early and late stages of the $P_2$ intermediate, in which only the latter leads to the fully-open channel pore (*Lórenz-Fonfría et al., 2015*). Furthermore, site-directed labeling with infrared sensitive, unnatural amino acids subsequently showed more specifically, that espacially hydration changes in the intracellular gate are correlated with channel opening (*Krause et al., 2019*). In summary, our TR-SFX experiment has revealed two critical conformational changes in TM7 and TM3 that occur during the early part of the photocycle, and these triggers would induce water and proton influx and subsequent pore dilation, although only slight structural perturbations in the gate constrictions were observed in the current study, as they were probably affected by the crystal packing interactions (*Figure 4—figure supplement 5*).

## Discussion

### Channel pore gating in ChR

The current results highlight the diversity and similarity of rhodopsin proteins. TM7 is covalently attached to the retinal and thus undergoes a similar shift in other rhodopsin proteins, such as the proton pump BR (*Nango et al., 2016*), and the Na⁺ pump KR2 (*Skopintsev et al., 2020*), albeit with some variations. Therefore, this structural movement is probably conserved in rhodopsin proteins. Most notably, the retinal kink and TM3 shift were also observed in KR2, even though it is evolutionally and functionally distant from ChRs, while such a conformational change was not observed in BR. In BR, the retinal is transiently kinked toward the opposite direction, but only during the very early time after photoexcitation (around 10 ps to 16 ns), and causes totally different changes in the protein part. Therefore, the retinal kink and subsequent conformational changes are likely to be affected by the residues constituting the retinal pocket. Indeed, the outward shift of TM3 was not observed in the Cys167 mutant of C1C2, as elucidated by the light-activated projection maps on the 2D

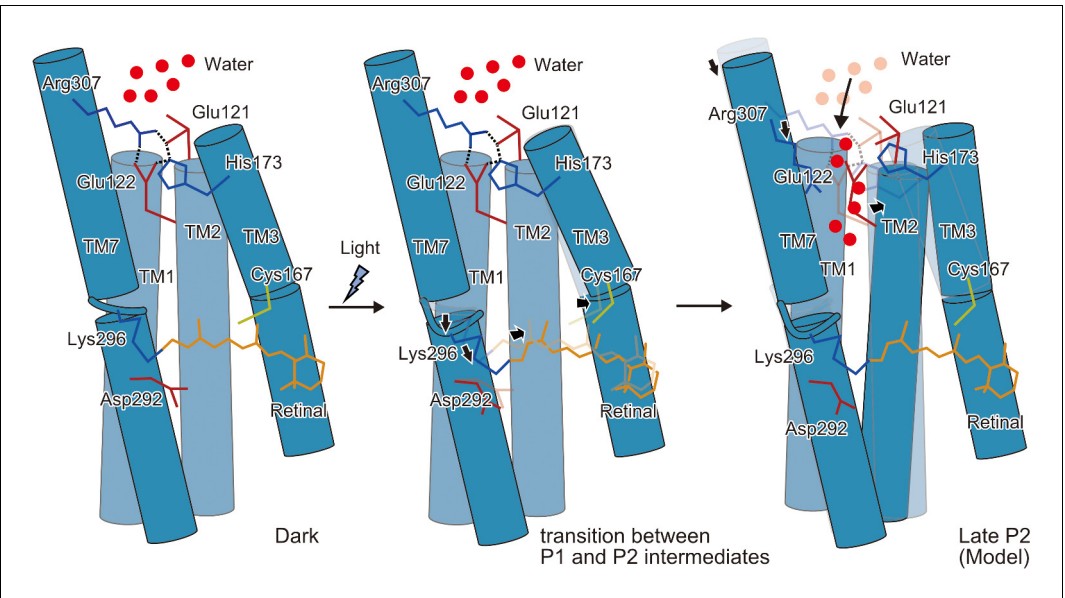

**Figure 6.** Schematic model of the C1C2 channel opening. The inner gate formed by Glu121, Glu122, His173, and Arg307 prevents water influx in the initial dark-adapted state. After the retinal isomerization reaction, the retinal twists and shifts toward Cys167 and induces an outward shift of TM3, originating at the kink introduced by Pro168. Isomerization of the retinal also induces a downward shift of the middle portion of TM7, especially at the retinal-attached residue, Lys296 (transition between the P1 and P2 intermediates). These movements are likely to induce water influx, by destabilizing the intracellular constriction, and further induce rearrangement of the surrounding helices (Open model).

crystallographic study (*Müller et al., 2015*). A recent MD simulation demonstrated the hydrogen bonding interactions between the deprotonated Schiff base and the hydroxyl group of Thr166, located beside the Cys167 residue (*Cheng et al., 2018*), which are likely to affect the retinal conformation. The same threonine is highly conserved among microbial rhodopsins, and in *Cr*ChR2 it was also proposed to couple the protonation changes in the counterion region to the hydrogen bonding changes in the DC-pair and the conformational changes of TM3 (*Watanabe et al., 2012*). Together, the hydrogen-bonding network beside the retinal, involving the DC-pair, may affect the retinal conformation after isomerization, in good agreement with the experimental evidence that the residues lining the retinal pocket affect channel kinetics (*Oda et al., 2018*). These results would pave the way for the development of novel optogenetic tools with different kinetics.

## Materials and methods

### Laser flash photolysis experiment

The time-evolution of the transient absorption changes of photo-excited C1C2 was measured as previously described (*Inoue et al., 2013*; *Inoue et al., 2016*). The purified protein sample was solubilized in buffer containing 150 mM NaCl, 50 mM Tris-HCl (pH 8.0), 0.05% n-dodecyl-β-D-maltoside (DDM), 5% glycerol, and 0.01% cholesteryl hemisuccinate (CHS). For the measurement of C1C2 in liposomes, the purified C1C2 protein was mixed with a lipid mixture of POPE and POPG (molar ratio = 3: 1) at a protein-to-lipid molar ratio of 1: 50 (the final concentrations of C1C2 and lipids were adjusted to 5 μM and 250 μM, respectively), and the DDM was removed by BioBeads (SM-2, Bio-Rad). The sample solution was placed in a quartz cuvette with a 1 cm optical path length and illuminated with a 480 nm beam generated by an OPO system, pumped by the third harmonics generation of a nanosecond $Nd^{3+}$: YAG laser (LT-2214, LOTIS TII, Minsk, Republic of Belarus). The intensities of the transmitted probe light from a Xe arc lamp (L8004, Hamamatsu Photonics, Japan) were measured before and after laser excitation with an ICCD linear array detector (C8808-01, Hamamatsu, Japan) at 24°C, and transient absorption spectra were obtained by calculating the ratios

between them. Ninety spectra were averaged to improve the signal-to-noise (S/N) ratio. To obtain the detailed time evolution of the transient absorption change after photoexcitation, the change in the intensity of the monochromated output of a Xe arc lamp (L9289-01, Hamamatsu Photonics, Japan), passed through the sample at 24°C, was observed by a photomultiplier tube (R10699, Hamamatsu Photonics, Japan) at 375, 445, and 520 nm. The signal from the photomultiplier tube was averaged to improve the S/N ratio and stored in a digital-storage-oscilloscope (DPO7104, Tektronix, Japan). The obtained kinetics data were fitted with a multi-exponential function.

Dark adaptation kinetics of C1C2 was measured for the 0.05% DDM-solubilized sample at 24°C, as previously described (*Inoue et al., 2016*). The dark-adapted sample was illuminated for 1 min by using the output from a 1-kW tungsten–halogen projector lamp (Master HILUX-HR, Rikagaku, Japan), through an interference filter at 460 nm. After turning off the light, the spectra or the absorption at a specific wavelength were measured by a UV-visible spectrometer (V-730, JASCO).

## Flash photo-activation current measurement

HEK293 cells were cultured in Dulbecco's Modified Medium (Biochrom), supplemented with 10% (v/v) fetal bovine serum (FBS Superior; Biochrom), 1 µM all-*trans* retinal, and 100 µg/ml penicillin/streptomycin (Biochrom, Berlin, Germany). Cells were seeded on poly-lysine coated glass coverslips at a concentration of $1 \times 10^5$ cells/ml, and transiently transfected using the FuGENE HD Transfection Reagent (Promega) 28–48 hr before measurement. C1C2 was fused to an mCherry-fluorophor and expressed under the control of the CMV-promotor.

Patch pipettes were prepared from borosilicate glass capillaries (G150F-3; Warner Instruments) using a P-1000 micropipette puller (Sutter Instruments), and fire polished with a final pipette resistance of 1.5–3.0 MΩ. Fluorescent cells were identified using an Axiovert 100 inverted microscope (Carl Zeiss). Single ns laser pulses were generated by an Opolette Nd:YAG laser/OPO system (OPO-TEK), selected by an LS6ZM2 shutter system (Vincent Associates), and delivered to the cells through the optics of the microscope and a 90/10 beamsplitter (Chroma). The laser intensity was set to 5% using the built-in motorized variable attenuator, resulting in a pulse energy of $100 \pm 20$ µJ/mm². Photocurrents were recorded with an AxoPatch 200B amplifier (Molecular Devices) at a maximal bandwidth of 100 kHz and digitized using a DigiData 1440A digitizer (Molecular Devices) at a sampling rate of 250 kHz. Membrane resistance was typically >1 GΩ, while access resistance was below 10 MΩ. Pipette capacity, series resistance, and cell capacity compensations were applied. A 140 mM NaCl agar bridge was employed for the reference bath electrode. Intracellular pipette and extracellular bath solutions contained 110 mM NaCl, 1 mM KCl, 1 mM CsCl, 2 mM $CaCl_2$, and 2 mM $MgCl_2$, and were buffered to pH 6.0, 7.2, or 8.0 by 10 mM MES, HEPES, or TRIS (with glucose added to 290 mOsm for intracellular and 310 mOsm for extracellular solutions). Measurements were controlled by the pCLAMP software (Molecular Devices). Post measurement photocurrent recordings were first baseline corrected and time shifted to align with the rising edges of the Q-switch signals of the laser pulse, using the Clampfit 10.4 software (Molecular Devices). They were then binned to 50 logarithmically spaced data points per temporal decade, normalized to peak photocurrents at −60 mV, and averaged for up to 10 individual repetitions for each cell and voltage condition with a custom-written Matlab script (MathWorks). Kinetic analysis was performed in Origin 9.1 (OriginLab). Passive photocurrents were calculated by subtracting the photocurrents at 0 mV, which were recorded in the absence of an electrochemical gradient and represent the inner protein charge transfer reactions. For sufficient statistical significance, each measurement was repeated multiple times on different biological replicates, in at least three independent experiments. The exact number of biological replicates for each measurement is provided in the figure legend.

## Time-resolved visible absorption spectroscopy

The photocycle reaction was induced by a 6-ns, 470 nm pump pulse from an optical parametric oscillator (OPO) (NT230, EKSPLA), and the spectral changes during the reaction were measured by using a fiber-coupled spectrometer (Flame-S, Ocean Optics), with a microsecond white-light probe pulse from a Xe-flash lamp (L11316-11-11, Hamamatsu Photonics). A portion of the probe light (5%) was used to correct the pulse-to-pulse fluctuation of the probe light intensity. The pump pulse and the probe light were focused on the sample point, with beam diameters of 300 µm and 40 µm, respectively. The pump energy was reduced to 15 µJ (0.21 nJ/µm [*Schneider et al., 2015*]), to avoid

sample damage by the repetitive pump illuminations during the data accumulation. The pump pulse, the probe light source, and the spectrometer were synchronized by pulse generators (DG645, Stanford Research Systems), and the pump-probe delay time was adjusted with a timing jitter of ±20 ns. The repetition rates of the pump and the probe were 0.5 Hz and 1 Hz, respectively. In the solution measurement, the sample was packed in a spinning quartz cell with an optical path length of 1 mm and spun at 50 rpm during the measurement. The sample concentration was ~10 mg/ml, dissolved in 50 mM Tris-HCl, pH 8.0, containing 150 mM NaCl and 5% glycerol. In the microcrystal measurement, we used the C1C2 crystals obtained under similar conditions as in the TR-SFX (below). Specifically, a pillar of 30 µl protein-laden LCP, approximately 2 cm in length, was extruded from the syringe and soaked in 900 µl of the crystallization buffer, containing 100 mM HEPES, pH 7.0, 280 mM $KH_2PO_4$, and 30% PEG500DM, and incubated for 1 week in the dark at 20°C, which yielded crystals in a liquid sponge phase floating on the buffer solution. The sponge phase layers were transferred by pipetting and packed in two quartz windows with a 50 µm-thick spacer. Only a few absorption signals from the non-crystallized area were observed, and we used the transmitted light intensity from the non-crystallized area as the reference to calculate the absorption spectra of C1C2. The sample temperature was kept at 20°C for both the solution and microcrystal measurements. To decrease the noise level of the time-resolved spectra, singular value decomposition (SVD) (*Shrager and Hendler, 1998*) was used with Igor Pro (Wave Matrix). In the SVD, matrix $A$ comprising the intensity changes at each wavelength and time (column and row, respectively) was decomposed according to $A = USV^T$, where $S$ represents the diagonal matrix with the singular values in decreasing order. This allows an evaluation of the number of significant orthonormal basis vectors for the wavelength (columns of $U$) and the time (columns of $V$). By reconstructing the matrix $A'$ from only the most significant components, random noise was eliminated from the raw spectra. The time-resolved spectra for the microcrystal and solution samples were well reconstructed by two and three components, respectively. Based on the SVD, global fitting was then performed for the time-resolved reconstructed spectra with Igor Pro (Wave Matrix). In the global fitting, the spectra were fitted with a sum of exponentials with apparent time constants $\tau_i$ as global parameters, where $i$ is the number of components used for the spectral reconstruction in the SVD analysis.

## Crystallization of C1C2

The vector construction and protein production and purification were performed according to the previous report (*Kato et al., 2012*). C1C2 was finally purified by size-exclusion chromatography, in 150 mM NaCl, 50 mM Tris-HCl, pH 8.0, 5% glycerol, 0.05% DDM, and 0.01% CHS, and concentrated to ~15 mg/ml for crystallization. C1C2 was mixed with monoolein (Sigma) in a 2:3 protein to lipid ratio (w/w), using a 100 µl volume Gastight syringe (Hamilton). A pillar of 30 µl protein-laden LCP, approximately 2 cm in length, was extruded from the syringe and soaked in 900 µl of the crystallization buffer, containing 100 mM MES, pH 6.9, 100 mM Na formate, and 30% PEG500DM, and incubated for 2–3 weeks in the dark at 20°C. Crystals were observed with a Hirox KH8700 digital microscope.

## Sample preparation for LCP-SFX

100 µl of LCP pillars with grown C1C2 microcrystals were collected and loaded into a Hamilton syringe and mixed with an equivalent volume of buffer-containing LCP (2:3 ratio of the buffer containing 150 mM NaCl and 50 mM Tris-HCl, pH 8.0, to monoolein). A 10 µl aliquot of liquid paraffin (Wako) was further added to the LCP material and homogenously mixed to enable smooth flow in the SFX experiment. All of these procedures were performed in the dark under red LED lights (over 600 nm wavelength).

## Diffraction experiment at SACLA-FEL

Single-shot XFEL data collection was performed, using femtosecond X-ray pulses from the SACLA at BL3. The pulse parameters of SACLA were as follows: pulse duration, <10 fs; X-ray energy, 7 keV; energy bandwidth, 33.5013 eV (FWHM); pulse flux, 2.0 × 1011 photon/pulse photons per pulse; beam size 1.5 µm (H) × 1.5 µm (W); repetition rate, 30 Hz. The C1C2 crystals were mixed with protein-less LCP and paraffin oil, loaded into an injector with a 75 µm inner diameter nozzle, and set in a diffraction chamber filled with helium gas, in a setup called Diverse Application Platform for Hard

X-ray Diffraction in SACLA (DAPHNIS). The flow rate was set to 2.5 µl/min (9.4 mm/s). The diffraction patterns were recorded on a multiport CCD detector with an eight sensor module (*Kameshima et al., 2014*). The excitation laser pulses were provided at 15 Hz, and the XFEL pulses had a repetition rate of 30 Hz. Each 'pump-on' image was followed by 'pump-off' images, which were recorded separately. The pump-on images were used to analyze the structural differences, whereas the diffraction data for the dark-state were collected in a separate run to avoid light contamination.

## Pump laser setup

To advance the C1C2 samples to the excited state, a 6 ns visible light pump pulse with a wavelength of 470 nm was provided to the samples at $\Delta t$ = 1 µs, 50 µs, 250 µs, 1 ms, and 4 ms, from an OPO (NT230, EKSPLA), which was Q-switched at 15 Hz using a pulse generator (DG645, Stanford Research Systems). The pulse generator was also used to synchronize the pump pulse with the XFEL pulse and control the delay time ($\Delta t$) with a timing jitter of <1 ns. Samples were thus irradiated from both sides of the LCP microjet, a geometry that was chosen to optimize the crystal excitation efficiency. The pump focal size was set to 40 µm (FWHM) and the pump energy was 20 µJ (10 µJ from each direction) at the sample point. The pump beam center was aligned 6 µm upstream from the XFEL spot. The pump light was scattered, as the LCP microjet is semi-transparent. To avoid the interference from the pump scattering, the sample flow rate (9.4 mm/s) was set to be the same as that in the TR-SFX analysis of BR, in which each pump-illuminated crystal for the pump-on XFEL images was separated by 630 µm. A portion of the pump beam (5%) was isolated with a beam sampler, and its photodiode-detected signal was used to monitor the laser status and to tag the diffraction image with 'light-on'. Since the repetition rate of the pump laser was half of that of the XFEL, the 'light-on' data and the 'dark' data were collected alternately.

## Data processing and structural analysis

Data collection was monitored by a real-time data processing pipeline (*Nakane et al., 2016*), based on Cheetah (*Barty et al., 2014*). Diffraction images with more than 20 spots were considered as hits and processed in CrystFEL, version 0.6.3 (*White et al., 2012*), with the parameters of hitfinderAlgorithm = 6, hitfinderADC = 40, hitfinderMinSNR = 5. Diffraction spots for indexing were detected by the built-in spot finding algorithms 'zaef' (*Zaefferer, 2000*) or 'peakfinder8' (*Barty et al., 2014*) in CrystFEL, and indexing was performed with DirAx (*Duisenberg, 1992*). Intensities were merged by Monte Carlo integration with the process_hkl command in the CrystFEL suite, with the per-image resolution cutoff (–push-res 1.2). Data collection statistics are summarized in *Supplementary file 1*. As the cell parameters do not vary among the different delayed-time points, the difference Fourier electron density maps ($|F_{obs}|^{light} - |F_{obs}|^{dark}$)·$\exp[i\Phi_{calc}]$ were calculated by FFT (*Ten Eyck, 1973*) in the CCP4 suite (*Winn et al., 2011*), with phases calculated from the refined dark state model.

## Structural refinement

The structure was solved by the molecular replacement method implemented in Phenix Phaser-MR, using the model of C1C2 (PDB: 3UG9). The structure was manually modified to fit into the electron density maps, using the program Coot (*Emsley et al., 2010*), and then refined with the programs Phenix (*Adams et al., 2010*) and Refmac5 in the CCP4 suite (*Vagin et al., 2004*). The model structures of the respective delayed time points were generated by real-space refinement against the difference-Fourier maps by COOT, with the restraints of protein geometry and retinal planarity between $C_5$ and $C_{13}$. Extrapolated data were calculated by a linear approximation, as follows: $F_{Extra}$ = ($F_o^{light} - F_o^{dark}$)/R + $F_o^{dark}$, where R represents the activation ratio of molecules and $F_{Extra}$ represents the extrapolated structure factor amplitude. $F_o^{light}$ was scaled to $F_o^{dark}$ by using the scaleit program in the CCP4 suite (*Howell and Smith, 1992*) before calculating $F_{Extra}$. The activation ratio R (the ratio of molecules in the excited state after light illumination) was determined by calculating extrapolated maps with phases of the dark state at different activation ratios in steps of 0.01, until a feature of the dark state at Cys167 disappeared. Next, 4 ms model was manually adjusted to fit the extrapolated map and refined by Refmac5 against $F_{Extra}$ and its estimated error. Then, each time point model was refined by Refmac5 with external structure restraints using 4 ms structure by the Geman–McClure

function (*Murshudov et al., 2011*) the refinement statistics are. Figures were prepared with Cuemol (http://www.cuemol.org).

## Acknowledgements

We thank Sae Okazaki and Masae Miyazaki for assistance in obtaining the C1C2 crystals, and Rie Tanaka, Toshi Arima, Ayumi Yamashita, and Jun Kobayashi for assistance in the collection of the X-ray data diffraction data at SACLA. We also thank the beamline staff scientists at SACLA and BL32XU of SPring-8 (Hyogo, Japan). The XFEL experiments were performed at the BL2 of SACLA, with the approval of the Japan Synchrotron Radiation Research Institute (JASRI) (proposal nos. 2015B8053, 2016A8054, 2016B8067, 2017A8022, and 2017B8035), and SPring-8 BL32XU (20170511, 20170721, and 20170728), with the approval of RIKEN. We acknowledge computational support from the SACLA HPC system and Mini-K super computer system.

## Additional information

### Funding

| Funder | Grant reference number | Author |
| --- | --- | --- |
| Japan Society for the Promotion of Science | 16H06294 | Osamu Nureki |
| Japan Society for the Promotion of Science | 18J21256 | Kazumasa Oda |
| Japan Society for the Promotion of Science | 17H05000 | Tomohiro Nishizawa |
| Japan Science and Technology Agency | JPMJPR14L9 | Kunio Hirata |
| Japan Science and Technology Agency | JPMJPR14L8 | Tomohiro Nishizawa |
| Japan Agency for Medical Research and Development | Platform Project for Supporting Drug Discovery andLife Science Research (Basis for Supporting Innovative Drug Discovery and Life Science Research (BINDS)) | Kunio Hirata |
| Deutsche Forschungsgemeinschaft | SFB1078 (B2) | Johannes Vierock |
| Cluster of Excellence in Inflammation Research | UniCat | Johannes Vierock |
| Cluster of Excellence in Inflammation Research | BIG-NSE | Johannes Vierock |
| Cluster of Excellence in Inflammation Research | E4 | Peter Hegemann |
| Cluster of Excellence in Inflammation Research | SPP 1926 | Johannes Vierock |
| European Research Council | ERC-2016-StG 714762 | Johannes Vierock |
| Hertie Foundation | Senior Research Professor | Peter Hegemann |
| Ministry of Education, Culture, Sports, Science and Technology | X-ray Free-Electron Laser Priority Strategy Program | So Iwata |

The funders had no role in study design, data collection and interpretation, or the decision to submit the work for publication.

## Author contributions
Kazumasa Oda, Conceptualization, Data curation, Investigation, Visualization, Methodology, Writing - original draft, Project administration, Writing - review and editing; Takashi Nomura, Conceptualization, Data curation, Investigation, Methodology, Writing - original draft, Writing - review and editing; Takanori Nakane, Data curation, Software, Supervision; Keitaro Yamashita, Software and model refinement; Keiichi Inoue, Data curation, Validation, Investigation, Writing - original draft, Writing - review and editing; Shota Ito, Kunio Hirata, Kota Katayama, Tatsuya Ikuta, Itsuki Ishigami, Tamaki Izume, Rie Umeda, Ryuun Eguma, Satomi Oishi, Go Kasuya, Takafumi Kato, Tsukasa Kusakizako, Wataru Shihoya, Hiroto Shimada, Tomoyuki Takatsuji, Mizuki Takemoto, Reiya Taniguchi, Atsuhiro Tomita, Ryoki Nakamura, Masahiro Fukuda, Hirotake Miyauchi, Yongchan Lee, Tomoyuki Tanaka, Tetsunari Kimura, Tatsuro Shimamura, Takaaki Fujiwara, Yasuaki Yamanaka, Investigation; Johannes Vierock, Data curation, Formal analysis, Visualization, Writing - original draft, Writing - review and editing; Andrés D Maturana, Data curation, Investigation; Eriko Nango, Conceptualization, Supervision, Investigation; Rie Tanaka, Ryuichiro Ishitani, So Iwata, Supervision, Project administration; Michihiro Sugahara, Investigation, Project administration; Shigeki Owada, Kensuke Tono, Software, Supervision; Yasumasa Joti, Software, Supervision, Investigation; Shigehiko Hayashi, Supervision; Hideki Kandori, Supervision, Writing - review and editing; Peter Hegemann, Supervision, Validation, Visualization, Writing - original draft, Writing - review and editing; Minoru Kubo, Data curation, Supervision, Validation, Investigation, Visualization, Writing - original draft, Project administration, Writing - review and editing; Tomohiro Nishizawa, Conceptualization, Data curation, Supervision, Validation, Investigation, Visualization, Writing - original draft, Project administration, Writing - review and editing; Osamu Nureki, Conceptualization, Funding acquisition, Project administration, Writing - review and editing

## Author ORCIDs
Kazumasa Oda (ID) https://orcid.org/0000-0003-0996-0393
Takanori Nakane (ID) https://orcid.org/0000-0003-2697-2767
Keiichi Inoue (ID) https://orcid.org/0000-0002-6898-4347
Johannes Vierock (ID) https://orcid.org/0000-0001-7368-5539
Go Kasuya (ID) https://orcid.org/0000-0003-1756-5764
Mizuki Takemoto (ID) https://orcid.org/0000-0002-6339-6431
Eriko Nango (ID) https://orcid.org/0000-0001-9851-7355
Ryuichiro Ishitani (ID) http://orcid.org/0000-0002-4136-5685
Peter Hegemann (ID) http://orcid.org/0000-0003-3589-6452
Tomohiro Nishizawa (ID) https://orcid.org/0000-0001-7463-8398
Osamu Nureki (ID) https://orcid.org/0000-0003-1813-7008

## Decision letter and Author response
Decision letter https://doi.org/10.7554/eLife.62389.sa1
Author response https://doi.org/10.7554/eLife.62389.sa2

# Additional files

## Supplementary files
• Supplementary file 1. Crystallographic data and refinement statistics. Values in parenthesis are those of the highest resolution shell.

• Supplementary file 2. Refinement statistics associated with extrapolated data. Because of the large variance of the extrapolated structure factor amplitude, the R values tend to be worse.

• Transparent reporting form

## Data availability
Coordinates and structure factors for the TR-SFX structures have been deposited in the PDB with the accession codes 7C86 (dark), 7E6Y (1 μs), 7E6Z (50 μs), 7E70 (250 μs), 7E71 (1 ms), and 7E6X(4

ms). Raw diffraction images have been deposited in the Coherent X-ray Imaging Data Bank (accession ID 150, https://doi.org/10.11577/1768368).

The following datasets were generated:

| Author(s) | Year | Dataset title | Dataset URL | Database and Identifier |
|---|---|---|---|---|
| Oda K, Nomura T, Nakane T, Yamashita K, Inoue K, Ito S, Vierock J, Hirata K, Maturana AsD, Katayama K, Ikuta T, Ishigami I, Izume T, Umeda R, Eguma R, Oishi S, Kasuya G, Kato T, Kusakizako T, Shihoya W, Shimada H, Takatsuji T, Takemoto M, Taniguchi R, Tomita A, Nakamura R, Fukuda M, Miyauchi H, Lee Y, Nango E, Tanaka T, Tanaka R, Sugahara M, Kimura T, Shimamura T, Fujiwara T, Yamanaka Y, Owada S, joti, Joti Y, Tono K, Ishitani R, Hayashi S, Kandori H, Hegemann P, Iwata S, Kubo M, Nishizawa T, Nureki O | 2021 | Time-resolved serial femtosecond crystallography reveals early structural changes in channelrhodopsin: Dark state structure | https://www.rcsb.org/structure/7C86 | RCSB Protein Data Bank, 7C86 |
| Oda K, Nomura T, Nakane T, Yamashita K, Inoue K, Ito S, Vierock J, Hirata K, Maturana AsD, Katayama K, Ikuta T, Ishigami I, Izume T, Umeda R, Eguma R, Oishi S, Kasuya G, Kato T, Kusakizako T, Shihoya W, Shimada H, Takatsuji T, Takemoto M, Taniguchi R, Tomita A, nakamura, Fukuda M, Miyauchi H, Lee Y, Nango E, Tanaka T, Tanaka R, Sugahara M, Kimura T, Shimamura T, Fujiwara T, Yamanaka Y, Owada S, Joti Y, Tono K, Ishitani R, Hayashi S, Kandori H, Hegemann P, Iwata S, Kubo M, Nishizawa T, Nureki O | 2021 | Time-resolved serial femtosecond crystallography reveals early structural changes in channelrhodopsin | https://dx.doi.org/10.11577/1768368 | Coherent X-ray Imaging Data Bank, 10.11577/1768368 |
| Oda K, Nomura T, Nakane T, Yamashita K, Inoue K, Ito S, Vierock J, Hirata K, Maturana | 2021 | Time-resolved serial femtosecond crystallography reveals early structural changes in channelrhodopsin: 1 µs structure | https://doi.org/10.2210/pdb7E6Y/pdb | Worldwide Protein Data Bank, 10.2210/pdb7E6Y/pdb |

| | | | | |
|---|---|---|---|---|
| AsD, Katayama K, Ikuta T, Ishigami I, Izume T, Umeda R, Eguma R, Oishi S, Kasuya G, Kato T, Kusakizako T, Shihoya W, Shimada H, Takatsuji T, Takemoto M, Taniguchi R, Tomita A, Nakamura R, Fukuda M, Miyauchi H, Lee Y, Nango E, Tanaka T, Tanaka R, Sugahara M, Kimura T, Shimamura T, Fujiwara T, Yamanaka Y, Owada S, Joti Y, Tono K, Ishitani R, Hayashi S, Kandori H, Hegemann P, Iwata S, Kubo M, Nishizawa T, Nureki O | | | | |
| Oda K, Nomura T, Nakane T, Yamashita K, Inoue K, Ito S, Vierock J, Hirata K, Maturana AsD, Katayama K, Ikuta T, Ishigami I, Izume T, Umeda R, Eguma R, Oishi S, Kasuya G, Kato T, Kusakizako T, Shihoya W, Shimada H, Takatsuji T, Takemoto M, Taniguchi R, Tomita A, Nakamura R, Fukuda M, Miyauchi H, Lee Y, Nango E, Tanaka T, Tanaka R, Sugahara M, Kimura T, Shimamura T, Fujiwara T, Yamanaka Y, Owada S, Joti Y, Tono K, Ishitani R, Hayashi S, Kandori H, Hegemann P, Iwata S, Kubo M, Nishizawa T, Nureki O | 2021 | Time-resolved serial femtosecond crystallography reveals early structural changes in channelrhodopsin: 50 µs structure | https://doi.org/10.2210/pdb7E6Z/pdb | Worldwide Protein Data Bank, 10.2210/pdb7E6Z/pdb |
| Oda K, Nomura T, Nakane T, Yamashita K, Inoue K, Ito S, Vierock J, Hirata K, Maturana AsD, Katayama K, Ikuta T, Ishigami I, Izume T, Umeda R, Eguma R, Oishi S, Kasuya G, Kato T, Kusakizako T, Shihoya W, Shimada H, Takatsuji T, Takemoto M, Taniguchi R, Tomita A, Nakamura R, Fukuda M, | 2021 | Time-resolved serial femtosecond crystallography reveals early structural changes in channelrhodopsin: 250 µs structure | https://doi.org/10.2210/pdb7E70/pdb | Worldwide Protein Data Bank, 10.2210/pdb7E70/pdb |

| | | | | | |
|---|---|---|---|---|---|
| Miyauchi H, Lee Y, Nango E, Tanaka T, Tanaka R, Sugahara M, Kimura T, Shimamura T, Fujiwara T, Yamanaka Y, Owada S, Joti Y, Tono K, Ishitani R, Hayashi S, Kandori H, Hegemann P, Iwata S, Kubo M, Nishizawa T, Nureki O | | | | | |
| Oda K, Nomura T, Nakane T, Yamashita K, Inoue K, Ito S, Vierock J, Hirata K, Maturana AsD, Katayama K, Ikuta T, Ishigami I, Izume T, Umeda R, Eguma R, Oishi S, Kasuya G, Kato T, Kusakizako T, Shihoya W, Shimada H, Takatsuji T, Takemoto M, Taniguchi R, Tomita A, Nakamura R, Fukuda M, Miyauchi H, Lee Y, Nango E, Tanaka T, Tanaka R, Sugahara M, Kimura T, Shimamura T, Fujiwara T, Yamanaka Y, Owada S, Joti Y, Tono K, Ishitani R, Hayashi S, Kandori H, Hegemann P, Iwata S, Kubo M, Nishizawa T, Nureki O | 2021 | Time-resolved serial femtosecond crystallography reveals early structural changes in channelrhodopsin: 1 ms structure | https://doi.org/10.2210/pdb7E71/pdb | Worldwide Protein Data Bank, 10.2210/pdb7E71/pdb |
| Oda K, Nomura T, Nakane T, Yamashita K, Inoue K, Ito S, Vierock J, Hirata K, Maturana AsD, Katayama K, Ikuta T, Ishigami I, Izume T, Umeda R, Eguma R, Oishi S, Kasuya G, Kato T, Kusakizako T, Shihoya W, Shimada H, Takatsuji T, Takemoto M, Taniguchi R, Tomita A, Nakamura R, Fukuda M, Miyauchi H, Lee Y, Nango E, Tanaka T, Tanaka R, Sugahara M, Kimura T, Shimamura T, Fujiwara T, Yamanaka Y, Owada S, Joti Y, Tono K, Ishitani R, | 2021 | Time-resolved serial femtosecond crystallography reveals early structural changes in channelrhodopsin: 4 ms structure | https://doi.org/10.2210/pdb7E6X/pdb | Worldwide Protein Data Bank, 10.2210/pdb7E6X/pdb |

Hayashi S, Kandori
H, Hegemann P,
Iwata S, Kubo M,
Nishizawa T, Nureki
O

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
