## [Decision Letter]

**Acceptance summary:**

This paper describes the use of time-resolved serial femtosecond crystallography to investigate light-induced changes in the conformation of channelrhodopsin. The authors identified initial conformational changes that occur upon illumination, including a shift in the position of retinal as well as additional changes in the conformation of transmembrane helices 3 and 7. Using these results, the authors propose a model for how initial conformational changes may culminate in channel opening. This work advances our understanding of the molecular mechanisms of channelrhodopsin activation and of light-induced conformational change more generally.

**Decision letter after peer review:**

Thank you for submitting your article "Time-resolved serial femtosecond crystallography reveals early structural changes in channelrhodopsin" for consideration by *eLife*. Your article has been reviewed by three peer reviewers, and the evaluation has been overseen by a Reviewing Editor and Kenton Swartz as the Senior Editor. The following individual involved in review of your submission has agreed to reveal their identity: Doeke Hekstra (Reviewer #1).

The reviewers have discussed the reviews with one another and the Reviewing Editor has drafted this decision to help you prepare a revised submission.

As the editors have judged that your manuscript is of interest but revised data analysis is required before it is published, we would like to draw your attention to changes in our revision policy that we have made in response to COVID-19 (https://elifesciences.org/articles/57162). First, because many researchers have temporarily lost access to the labs, we will give authors as much time as they need to submit revised manuscripts. We are also offering, if you choose, to post the manuscript to bioRxiv (if it is not already there) along with this decision letter and a formal designation that the manuscript is "in revision at *eLife*". Please let us know if you would like to pursue this option. (If your work is more suitable for medRxiv, you will need to post the preprint yourself, as the mechanisms for us to do so are still in development.)

Summary:

This manuscript by K. Oda et al. describes the use of time-resolved serial femtosecond crystallography to investigate light-induced changes in the conformation of channelrhodopsin. The manuscript identifies initial conformational changes that occur upon illumination, including a shift in the position of retinal as well as additional changes in the conformation of transmembrane helices 3 and 7. The authors propose a model for how these initial changes may culminate in channel opening.

Essential revisions:

Several technical points were raised during the review process, but the most significant concern centers on the approach to model refinement. The description of the refinement process in the Materials and methods section is very brief, and is missing essential details such as the choice of restraints and the exact details of which difference maps were used. More generally, the use of real-space refinement directly against difference maps is not sufficiently described, and is potentially confounded by a variety of issues. For example, motion of an amino acid side chain into a region that would otherwise be occupied by an ordered water molecule could lead to little change in *F_o_-F_o_* density, or changes that are not readily interpretable. Reciprocal space refinement against extrapolated structure factors or reciprocal space refinement of partial occupancy models for each conformational state would be more appropriate (see below). At least one of these approaches should be included in a revised manuscript, along with appropriate statistical benchmarks such as real or reciprocal space correlation coefficients or R-factors. The reviewers also wish to emphasize that it is more appropriate to refine against structure factors corresponding to a conventional electron density (2*F_o_-F_c_* type) map than to a difference map. A second significant concern is the mismatch between the timescales of the QM/MM simulations and the experiments, which is not adequately explained. There are a variety of smaller technical points raised, which are discussed in detail below.

1) The authors mentioned low quantum efficiency of the retinal isomerization (~30% in C1C2), and then they built models onto the observed peaks in the difference maps. This can be problematic without validation, and the difference map signals can be misleading. In order to justify the correctness of the refined structures, the difference maps calculated between structures of light and dark states should be presented. Furthermore, more details should be included in structure refinement section, such as restraints, weighting factors, and validation metrics. Alternatively, the extrapolation approach has been adapted by several recent TR-SFX studies, such as the bR or KR2 studies (Science 365, 61-65, 2019); Nature 583, 314-318, 2020). Authors may want to try these approaches. In principle, it may also be possible to refine models of different states with partial occupancy based on the known photoconversion efficiency, which would also provide useful refinement statistics to assess model quality.

2) A vexing complication of the experiments is that the spectroscopic kinetics in crystal form differ markedly from those in solution. Clearly, the crystallographic data show that retinal isomerization is induced in the crystal and leads to conformational change at nearby positions. Spectroscopically, only two states are clearly distinguishable within the time scales of interest (P1/3 versus P2), reflecting the protonation state of the Schiff base. The authors speculate that they see a longer-lived P2 state in the crystal, apparently concurrent with a P1 or P3 state. This is reasonable, but it is not clear whether the authors really see a "P1 to P2 transition" (inclusion of shorter pump-probe delays might have been more convincing). The spectroscopic data likely miss detail of the structural transitions following photoexcitation and do not shed much light on how conformational dynamics may differ between crystal and solution.

3) The comparison between refined structures and the QM/MM simulation model is not convincing. QM/MM calculation results are based on short simulations. The timescales do not match with the ones studied in this TR-SFX. It is hard to make a fair comparison.

---

## [Author Response]

Essential revisions:Several technical points were raised during the review process, but the most significant concern centers on the approach to model refinement. The description of the refinement process in the Materials and methods section is very brief, and is missing essential details such as the choice of restraints and the exact details of which difference maps were used. More generally, the use of real-space refinement directly against difference maps is not sufficiently described, and is potentially confounded by a variety of issues. For example, motion of an amino acid side chain into a region that would otherwise be occupied by an ordered water molecule could lead to little change in F_o_-F_o_ density, or changes that are not readily interpretable. Reciprocal space refinement against extrapolated structure factors or reciprocal space refinement of partial occupancy models for each conformational state would be more appropriate (see below). At least one of these approaches should be included in a revised manuscript, along with appropriate statistical benchmarks such as real or reciprocal space correlation coefficients or R-factors. The reviewers also wish to emphasize that it is more appropriate to refine against structure factors corresponding to a conventional electron density (2F_o_-F_c_ type) map than to a difference map. A second significant concern is the mismatch between the timescales of the QM/MM simulations and the experiments, which is not adequately explained. There are a variety of smaller technical points raised, which are discussed in detail below.

According to the comments, we performed the reciprocal space refinement against the extrapolated structure factors, which is included below as a response to the reviewers.

1) The authors mentioned low quantum efficiency of the retinal isomerization (~30% in C1C2), and then they built models onto the observed peaks in the difference maps. This can be problematic without validation, and the difference map signals can be misleading. In order to justify the correctness of the refined structures, the difference maps calculated between structures of light and dark states should be presented. Furthermore, more details should be included in structure refinement section, such as restraints, weighting factors, and validation metrics. Alternatively, the extrapolation approach has been adapted by several recent TR-SFX studies, such as the bR or KR2 studies (Science 365, 61-65, 2019); Nature 583, 314-318, 2020). Authors may want to try these approaches. In principle, it may also be possible to refine models of different states with partial occupancy based on the known photoconversion efficiency, which would also provide useful refinement statistics to assess model quality.

We appreciate the reviewer’s critical comments on our study. We calculated the extrapolation maps according to the recent TR-SFX papers (Figure 4—figure supplement 3). First, we roughly estimated the ratio of the excited state by calculating extrapolated maps with different activation ratios until a feature of the dark state at Cys167 disappeared, and then modeled the excited states and performed the reciprocal refinement against the extrapolated maps. The obtained models confirmed the structural changes in TM3 and TM7, one of the major issues in the current manuscript (Figure 4—figure supplements 4, 5). However, this approach could not provide detailed information about the retinal conformation, because it showed only a weak density around the C_13_-C_14_ double bond of the retinal, probably due to its intrinsic flexibility (Figure 4—figure supplements 4, 5). Despite the lack of strong evidence, we speculate that the retinal adopts a kink toward TM3, based on the difference Fourier map that showed strong positive density beside the retinal, as well as the results of the QM/MM simulation. It should also be noted that a similar retinal kink and TM3 shift were visualized in recent work on another rhodopsin protein, KR2 (Skopintsev et al., 2020). Therefore, we supposed that the series of conformational changes is somehow conserved in the rhodopsin family proteins, although it was not observed in the representative member, bacteriorhodopsin. We would like to include both the normal difference Fourier maps (Figure 4) and the extrapolated maps (Figure 4—figure supplement 3), because high resolution diffraction data are required to calculate reliable extrapolation maps, and our maps thus contain a lot of noise.

2) A vexing complication of the experiments is that the spectroscopic kinetics in crystal form differ markedly from those in solution. Clearly, the crystallographic data show that retinal isomerization is induced in the crystal and leads to conformational change at nearby positions. Spectroscopically, only two states are clearly distinguishable within the time scales of interest (P1/3 versus P2), reflecting the protonation state of the Schiff base. The authors speculate that they see a longer-lived P2 state in the crystal, apparently concurrent with a P1 or P3 state. This is reasonable, but it is not clear whether the authors really see a "P1 to P2 transition" (inclusion of shorter pump-probe delays might have been more convincing). The spectroscopic data likely miss detail of the structural transitions following photoexcitation and do not shed much light on how conformational dynamics may differ between crystal and solution.

Conformational transitions are generally slower in crystals than in solution, and our spectroscopic study showed the same trend in C1C2. Figure 2—figure supplement 3 shows clear evidence that the P1 formation is completed within 10 μs, and the subsequent P2 formation is most prominent in 10 msec in crystals, which is slower than the corresponding steps in solution. Also, the lack of the P3 formation is consistent with the idea that the later reaction is more likely to be affected by crystal packing interactions. This information is sufficient to support the proposal that the dominant changes along the TR-SFX (occurring between 10 μs – 2.1 ms) are derived from the conformational transition from P1 to P2, although slight contamination by other intermediates (such as P3) might exist. It would be informative if we could understand the details of the early changes (including the precise population of photo-intermediates) after photoexcitation, but this is technically difficult because the spectrum changes in the tiny crystals of C1C2 are very weak. In addition, the spectroscopic studies only provide information about the micro-environmental changes around the retinal Schiff base and do not directly correlate with the global conformational changes of the protein. The previous bR TR-SFX study also suggested the lack of later conformational changes in the crystals, and suggested that the local changes in the Schiff base and the global structural changes are not necessarily coincident, especially in the later reactions. Therefore, the conclusion in the current manuscript is minimally dependent on the spectroscopic study. The observed changes in TM3 may occur during the P1 to P2 transition, and probably more quickly in solution.

3) The comparison between refined structures and the QM/MM simulation model is not convincing. QM/MM calculation results are based on short simulations. The timescales do not match with the ones studied in this TR-SFX. It is hard to make a fair comparison.

The molecular dynamics simulations in the QM/MM free energy geometry optimization by Cheng et al., 2018, showed the precursor state of the channel opening (early P2 state) on the timescale of tens to hundreds of nanoseconds. Although this time scale is much longer than that of the conventional QM/MM simulations (typically tens of picoseconds; e.g., Ardevol and Hummer, PNAS 115, 3557 (2018)), it is still shorter than the experimental time scale of the pore formation, 2 ~10 ms (Lorenz-Fonfria et al., 2013). However, please note that the timescales of the QM/MM simulations do not directly represent the experimental timescale of the early P2 formation. Spectroscopic measurements showed that the time constant of the early P2 state formation coincides with that of the proton translocation from the protonated Schiff base to its counterion, glutamate (Lorenz-Fonfria et al., 2013), exhibiting a deuterium kinetic isotope effect (Resler et al., Biophys. J. 109, 287 (2015)). These studies indicate that the early P2 formation is kinetically governed by the proton translocation. In contrast, the QM/MM calculations simulated the conformational changes of the protein after the proton translocation event. The proton translocation has already been well characterized experimentally. Therefore, it was omitted in the simulation to enable the modeling of the conformational changes triggered by the proton translocation, at the unprecedentedly high level of theory achieved by the combination of highly accurate ab initio quantum chemistry. The key chemical feature revealed by the QM/MM simulations was the torsional motion from a locally twisted conformation in the Schiff base, due to the increased electronic double bond character upon the proton translocation. This can be properly described by the ab initio QM treatment of the QM/MM simulations and is in excellent agreement with the current TR-SFX observation. The molecular events described in the simulations are distinctly different from those in the experiments. Therefore, although the simulation and current TR-SFX both indicated the retinal kink upon the proton transfer, the timescales of these results do not exactly match. These points are clarified in the revised manuscript (although the QM/MM simulation focuses only on the molecular events after the Schiff base deprotonation and thus does not exactly match the experimental time scale.).